# MitoToxy assay: A novel cell-based method for the assessment of metabolic toxicity in a multiwell plate format using a lactate FRET nanosensor, Laconic

Yasna Contreras-Baeza[1], Sebastián Ceballo[1], Robinson Arce-Molina[1,2], Pamela Y. Sandoval[1], Karin Alegría[1], Luis Felipe Barros[1], Alejandro San Martín[1] *

**1** Centro de Estudios Científicos (CECs), Valdivia, Chile, **2** Universidad Austral de Chile (UACh), Valdivia, Chile

* aalejo@cecs.cl

## Abstract

Mitochondrial toxicity is a primary source of pre-clinical drug attrition, black box warning and post-market drug withdrawal. Methods that detect mitochondrial toxicity as early as possible during the drug development process are required. Here we introduce a new method for detecting mitochondrial toxicity based on MDA-MB-231 cells stably expressing the genetically encoded FRET lactate indicator, Laconic. The method takes advantage of the high cytosolic lactate accumulation observed during mitochondrial stress, regardless of the specific toxicity mechanism, explained by compensatory glycolytic activation. Using a standard multi-well plate reader, dose-response curve experiments allowed the sensitivity of the methodology to detect metabolic toxicity induced by classical mitochondrial toxicants. Suitability for high-throughput screening applications was evaluated resulting in a Z'-factor > 0.5 and CV% < 20 inter-assay variability. A pilot screening allowed sensitive detection of commercial drugs that were previously withdrawn from the market due to liver/cardiac toxicity issues, such as camptothecin, ciglitazone, troglitazone, rosiglitazone, and terfenadine, in ten minutes. We envisage that the availability of this technology, based on a fluorescent genetically encoded indicator, will allow direct assessment of mitochondrial metabolism, and will make the early detection of mitochondrial toxicity in the drug development process possible, saving time and resources.

## Introduction

High-throughput screening (HTS) is a fundamental step in the hierarchical and long drug development process. Discovering and developing a drug starts with libraries consisting of thousands of chemicals, which are screened for a target activity using *in vitro* assays. Hits are selected for subsequent stages, then tested for activity and toxicity in cells, tissues, animal models, followed by clinical trials. Most candidates fail in clinical trials, as they are found to be ineffective in a physiological context or produce unwanted side effects. Rate of failure of lead

**Data Availability Statement:** All relevant data are within the manuscript and its Supporting Information files.

**Funding:** This work was supported by: ASM, 11150930, CONICYT-FONDECYT Initiation into Research Program; LFB, 14IEAT-28662, CORFO project "High Technology Business Innovation". The Centro de Estudios Científicos (CECs) is funded by the Chilean Government through the Centers of Excellence Basal Financing Program of CONICYT. The funders had no role in study design, data collection and analysis, decision to publish, or preparation of the manuscript.

**Competing interests:** The authors have declared that no competing interests exist.

compounds during the drug development process is called attrition, which can be quantified giving a wide view about the whole efficacy of drug development chain.

Pre-clinical and clinical safety, pharmacokinetics or bioavailability issues, and rationalization of the company portfolio are the main sources of drug attrition [1]. However, toxicity-related side-effects are by far the most important source of attrition during pre-clinical stages, being responsible for 59% of failures [1]. For instance, a comprehensive study combined data on drug attrition from AstraZeneca, Eli Lilly, GlaxoSmithKline, and Pfizer, which revealed that during 2000 and 2010 a total of 211 from 356 lead compounds were rejected due to safety concerns in the pre-clinical stage [1]. Additionally, in clinical trials toxicity is still the main source of attrition in phase I [1] and during phase II and III, the safety issues are the second source of attrition after efficacy [2–4]. For instance, in phase II failures due to toxicity during 2008–2010 and 2011–2012 were 19% and 22%, respectively [4]. For phase III, in the same period, attrition was 21% and 35% [4]. Despite lower attrition during phase II and III in clinical trials, approximately a quarter of failures, the cost and time invested in each lead compound at that stage of development is around 10 years and hundreds of US million dollars in R&D investment. Attrition at later stages of the development chain produces a higher economic impact and is the most important determinant of overall efficiency. These data suggest that although minimizing safety-related attrition has been a significant area of investment across the industry, it remains a key area to be improved.

Toxicity issues do not only cause problems at pre-clinical and clinical stages. During commercialization, safety issues are the main responsible of post-market drug withdrawal and black box warnings. For instance, studies have shown that between 1994 and 2006 a total of 38 FDA-approved drugs underwent post-market drug withdrawal due to safety concerns [5]. Another study found 19 drugs were withdrawn between 2002–2011 in the European market due to adverse drug reactions or safety concerns [6] and a comprehensive study of worldwide post-market withdrawal between 1950 and 2014 found a total of 462 medical products that were withdrawn from the market [7]. All these studies showed that the principal target for toxicity was the liver [5–7]. Additionally, a total of 111 drugs were highlighted with black box warnings by the FDA from January 2008 through to June 2015, being death and cardiovascular risk the most commonly argued reasons [8]. At these stages drugs were widely commercialized, producing mortality and legal issues with consequent monetary compensation, producing heavy losses for the industry.

Mitochondria is the main source of toxicity issues during drug discovery campaigns and clinical trials [9–13]. For instance, more than 80 individual chemical identities that inhibit complex I of the electron transport chain (ETC) alone have been reported [9]. Potential mitochondrial targets of drug-induced toxicity are not only restricted to primary targets such as ETC or uncoupled mitochondrial oxidative phosphorylation, but also to secondary targets including molecular regulation, substrate availability, and protein trafficking [12]. Higher numbers are expected due to the highly diverse mitochondrial proteome and the complexity of mitochondrial physiology.

Current methodologies to evaluate mitochondria damage take advantage of: i) *in vitro*, *in cellulo*, *ex vivo* and *in vivo* systems; ii) samples such as isolated mitochondria, permeabilized cells, living cells, tissue and animal samples and iii) readout such as mitochondrial membrane potential (MMP), oxygen consumption rate (OCR), adenosine triphosphate (ATP), extracellular acidification rate (ECAR), reactive oxygen species (ROS), glutathione (GSH), and viability assays. Also, a computational method to predict *in vivo* toxicity was recently described [14]. The ideal method the pharmaceutical industry has to offer a balance between throughput and physiological relevance. For instance, *in vivo* models provide low throughput because the sample preparation for measurements are laborious, but the physiological relevance is high

because the whole animal presents intact short and long-distance interactions between molecules, cells, and organs that allow the testing of key parameter such as bioavailability and cellular toxicity of lead molecules. Additionally, *in vitro* methodologies such as isolated mitochondria offer high throughput with low physiological relevance because they are easy to set up in a multi-plate reader but lack cytosolic context. New technologies are needed to complement the current methodologies and achieve a balance between throughput and physiological relevance.

Genetically encoded fluorescent indicators provide a minimally invasive approach for metabolite detection in living systems [15, 16]. They are built by the fusion of a ligand-binding moiety from bacteria to a fluorescent protein förster resonance energy transfer (FRET) pair module. Binding of the test molecule, triggers a conformational change that affects the relative distance and/or orientation between the acceptor/donor fluorescent proteins, causing an increase or decrease in FRET efficiency [17]. These genetically-encoded indicators have been instrumental to decipher molecular mechanisms involved in fast modulation of oxidative and glycolytic metabolism in the brain [15, 18–23]. Being fluorescent and genetically-encoded, these tools have a great potential to develop cell-based high-throughput methods for the pharmaceutical industry [24–28]. Using a FRET-based lactate indicator Laconic [29], we have observed that the mitochondrial damage induced by sodium azide or physiological mitochondrial modulator signals such as nitric oxide, produces a fast and acute cytosolic lactate accumulation [21, 29]. A phenomenon explained primarily by a reduction of mitochondrial pyruvate consumption and secondary glycolysis activation [21, 30, 31]. Both effects converge into increased cytosolic pyruvate and NADH levels, displacing the chemical equilibrium of lactate dehydrogenase (LDH) to lactate production within seconds. Therefore, we hypothesized that cytosolic lactate is a fast and sensitive metabolic beacon of mitochondrial dysfunction.

This paper describes a cell-based method to detect metabolic toxicity by measuring cytosolic lactate accumulation induced by mitochondrial damage. To develop and implement the methodology, we generated a cancer cell line MDA-MB-231 that stably expresses Laconic. To assess the suitability for high-throughput screening applications, Z'-Factor and inter-plate variability were determined using classical mitochondrial toxicants. As a proof of the concept, 13 compounds were tested in a pilot screening. The method provided rapid detection in 10 minutes of a panel of classical mitochondrial toxicants and allow the detection of previously described toxic drugs in a 96-well format using standard multi-plate readers.

## Materials and methods

Standard reagents and inhibitors were acquired from Sigma and Tocris. Plasmid encoding FRET sensor Laconic is available from addgene (www.addgene.org). Ad-Laconic (serotype 5) was custom made by Vector Biolabs. LV_Laconic was in-house generated using Viral Power System (ThermoFisher). MDA-MB-231 cell line was acquired from ATCC (ATCC® CRM-HTB-26™).

Mitochondrial toxicant stocks were prepared as follow: Azide was dissolved in nano-pure water, antimycin in 0.16% ethanol, myxothiazol, oligomycin, and rotenone in 0.8% Dimethyl Sulfoxide (DMSO). The 13 compounds used in the pilot screening were dissolved in DMSO at a final concentration of 0.2%. Master stock of p-chloromercuribenzene sulfonate (pCMBS) at 250 mM (5000x) was prepared in 10 M NaOH.

### Primary astrocytes culture

All animal procedures for the *in vitro* experiments were approved by the Institutional Animal Care and Use Committee of the Centro de Estudios Científicos (CECs). Animals used

for primary cultures were mixed F1 male mice (C57BL/6J × CBA/J), kept in an animal room under SPF conditions at a room temperature of 20 ± 2˚C, in a 12/12-h light/dark cycle and with free access to water and food. Mixed cortical cultures of neuronal and glia cells (1 –to 3-day-old neonates) was cultured in Neurobasal medium (Gibco) supplemented with 5 mM glucose and B27. Cells were seeded on petri dishes and superfused with KRH HEPES pH 7.4. The lactate imaging experiments with single-cell resolution were performed using cultured astrocytes at 11–14 days post-dissection. Real time experiments were performed with KRH HEPES bathing solution containing (in mM) 136 NaCl, 3 KCl, 1.25 CaCl$_2$, 1.25 MgCl$_2$, 10 HEPES/Tris and pH 7.2 at 36˚C with 300 mOsm, without carbon sources for mitochondrial toxicant experiments and 2 mM glucose and 1 mM lactate for Warburg index determination. Experiments were performed in upright Olympus FV1000 confocal microscope equipped with a 20X water-immersion objective (numerical aperture, 1.0) and a 440 nm solid-state laser.

## Generation of a clonal Laconic cell line

Recombinant lentiviral particles for Laconic were produced by co-transfection of lentiviral vectors LV-Laconic and Viral Power, lentiviral packaging mix (ThermoFisher) in HEK293FT-cells. Viral supernatants were harvested 72 hours after transfection, filtered through 0.45μm units (Millipore), concentrated, aliquoted, and stored at -80˚C. Transductions with Laconic lentivirus (1.3x10$^5$ TU/ml) were performed at a multiplicity of infection (MOI) of 3 in MDA-MB-231 cells. The selection of stable sensor expression was established under selective pressure using Blasticidin at 5ug/ml for 2 weeks after transfection. Cell cultures were processed by fluorescence-activated cell sorting (FACS) in order to isolate clonal cell lines of MDA-MB-231 expressing Laconic. Highly fluorescent cells were sorted, and single cells were collected individually in a 96 well plate. Each clone was propagated and screened based on the expression of the functional Laconic sensor.

## MitoTox reporter cell culture and measurements

MitoTox reporter cell line was cultured in Leibowitz media (Gibco) supplemented with 10% of fetal bovine serum (FBS) at 37˚C at atmospheric CO$_2$ levels in a 100 mm petri dish. Once cells reached 100% confluency, they were trypsinized using 5% trypsin (Gibco) and washed in PBS. Viability was determine using trypan blue staining and 10,000 cells were seeded into each well of a 96 well plate. Experiments in 96 well format were performed once cells reached 100% confluency after 6 days, during the incubation media was not replaced. All the plates with signal-to-noise ration below 1.3 for monomeric Teal Fluorescent Protein (mTFP) and Venus channels were discarded. Background signal was determined with KRH buffer without cells. Experiments consisted in washing out the culture media using KRH HEPES buffer containing (mM) 136 NaCl, 3 KCl, 1.25 CaCl$_2$, 1.25 MgCl$_2$, 10 HEPES/Tris and pH 7.2 at 37˚C with 300 mOsm. Then 200 μl of KRH HEPES buffer without any carbon sources was added to each well and the baseline was measured (R$_{MIN}$), then 100 μl of buffer was replaced with mitochondrial toxicant/controls and signal was measured to obtain a stimulated response (R$_{MAX}$). In experiments were pCMBS was used to block lactate exit, the inhibitor was added simultaneously with the mitochondrial toxicants. Data was collected in triplicated after 5, 10, 30, and 60 minutes of incubation. All the experiments in 96 well plates were performed in an EnVision® multiplate reader (PerkinElmer). Each well was excited at 430 nm and the intensity of mTFP and Venus fluorescence emission were recorded at 485 nm and 528 nm, using a set of filters.

## Data analysis

To plot the data, we used a ratio between mTFP and Venus channels. This ratio is proportional to lactate levels, since its binding to the sensor produced a decrease in FRET efficiency. This ratio was expressed in two formulas described below:

$$Normalized\ \Delta R = R(\frac{mTFP}{Venus})/R(\frac{mTFP}{Venus})min$$

Where R(mTFP/Venus) min corresponded to the minimal change of mTFP/Venus ratio during the single-cell experiments. Results in 96 well plates experiments were expressed by the ΔR%. This parameter was obtained from the difference between the $R_{MAX}$ and $R_{MIN}$ from the same well (before-after experiment) as follow:

$$\Delta R\% = \left( \frac{R(\frac{mTFP}{Venus})max - R(\frac{mTFP}{Venus})min}{R(\frac{mTFP}{Venus})min} \right) \times 100$$

To validate the suitability of the assay for high-throughput screening a coefficient called Z'-factor was calculated [32]. This coefficient is reflective of the assay signal dynamic range and the data variation associated with the signal measurements. The Z'-factor is a dimensionless parameter and is calculated with the formula below:

$$Z' = 1 - \frac{3(\sigma p + \sigma n)}{\mu p - \mu n}$$

Where 3σp is 3 times the standard deviation of the positive control, 3σn is 3 standard deviations from the negative control, μp average signal of the positive control and μn average signal of the negative control. To calculate the Z'-Factor we used the values obtained from wells treated with pCMBS plus the solvent as a negative control and wells treated with classical mitochondrial toxicants and pCMBS as a positive control.

Data from pilot screening experiments were found to pass normality distribution and therefore statistical significance of the changes in ΔR was assessed using one-way ANOVA. Multiple comparisons versus control groups (Holm-Sidak method), Overall significance level P < 0,05.

## Results

### Mitochondrial toxicants produced an acute increase of intracellular lactate levels

The mitochondrial poison azide, a well know ETC inhibitor at the cytochrome c level, induced an acute increase of lactate accumulation in astrocytes, brain cells that are highly sensitive to mitochondrial toxicants [29]. This observation is explained by blockage of mitochondrial pyruvate consumption and secondary glycolytic activation. Thus, accumulated cytosolic pyruvate is converted to lactate producing its cytosolic accumulation within seconds. With the aim of testing if this observation occurs in general and if azide induced-lactate accumulation can be reproduced using another mitochondrial toxicant with a different molecular target, we performed single-cell experiments exposing primary culture of astrocytes from the mouse cortex expressing Laconic (**Fig 1A**) to different mitochondria inhibitors that block the ETC at different levels (**Fig 1C**). To take advantage of the maximal dynamic range of the lactate FRET sensor the baseline measurements were performed using buffer without carbon sources to force start the experiments with lactate levels below 100 μM. Treatment with rotenone a complex I inhibitor, antimycin a complex III inhibitor, azide a cytochrome c inhibitor, and oligomycin an ATPase blocker, induced an acute cytosolic lactate accumulation in astrocytes (**Fig 1B**).

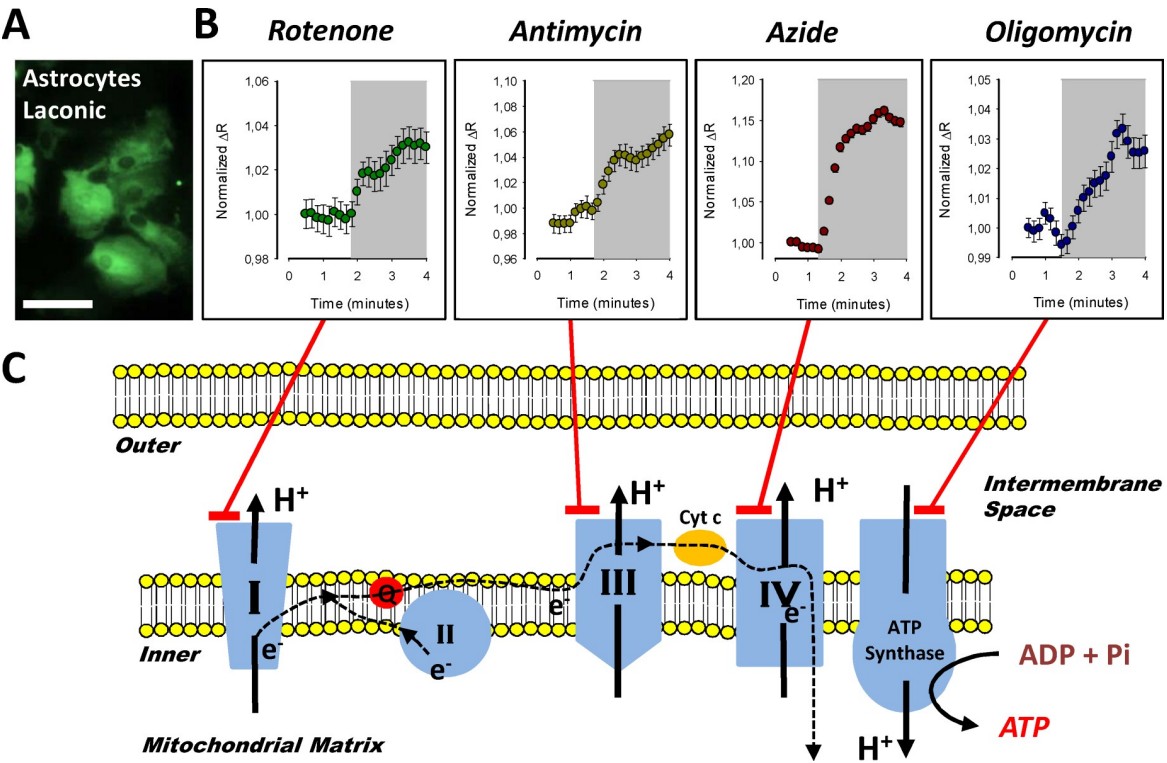

**Fig 1. Modulation of astrocytic lactate metabolism by mitochondrial toxicants.** A) Primary culture of cortical astrocytes from mice imaged at 440 nm excitation/535 nm emission. B) Cytosolic lactate accumulation induced by treatment with 32 μM rotenone, 16 μM of antimycin, 5 mM azide and 80 μM of oligomycin. Each data point is from an average of ten cells from three independent experiments. C) Schematic representation of molecular targets for each toxicant used in the primary culture of astrocytes.

Intracellular lactate levels rapidly increased and reached a new steady state in seconds, indicating fast mitochondrial dysfunction. Therefore, toxicity-induced lactate accumulation can be triggered by the blockage of ETC at different points. These observations suggest that intra-cellular lactate levels can be used as a wide specificity readout of mitochondrial dysfunction.

## Oxidative reporter cell line

Sensitive cell-based system to detect mitochondrial toxicity requires a highly activated oxidative metabolism. Primary cultures, in contrast to cell lines, are highly oxidative due to their non-cancer origin, therefore they are more sensitive to mitochondrial toxicants [33]. However, clonal cell lines offer advantages over primary cultures due to their HTS applications, such as low-cost production/maintenance, straightforward procedures for sample preparation, and they do not require the use of animals. In spite of the fact that immortalized cell lines generate most of their ATP from glycolysis [34], there are strategies to boost their oxidative metabolism. For instance: i) forcing cells to grow in media with galactose as the only carbon-source [35], ii) use growth media with oleic acid as the only carbon source [36] and iii) perform the experiments with cells cultured at a high confluence to increase cell-to-cell contact and to decrease the proliferative phenotype [36]. Therefore, we determined the magnitude of glycolytic and oxidative metabolism of a panel of cell lines and primary cultures of astrocytes using warburg index (WI) (**Fig 2A**). This parameter allows glycolytic and oxidative metabolism to be evaluated at single-cell resolution [29]. The protocol consists in the acute reversible perturbation of the intracellular lactate steady-state levels blocking the ETC with azide and then promoting its

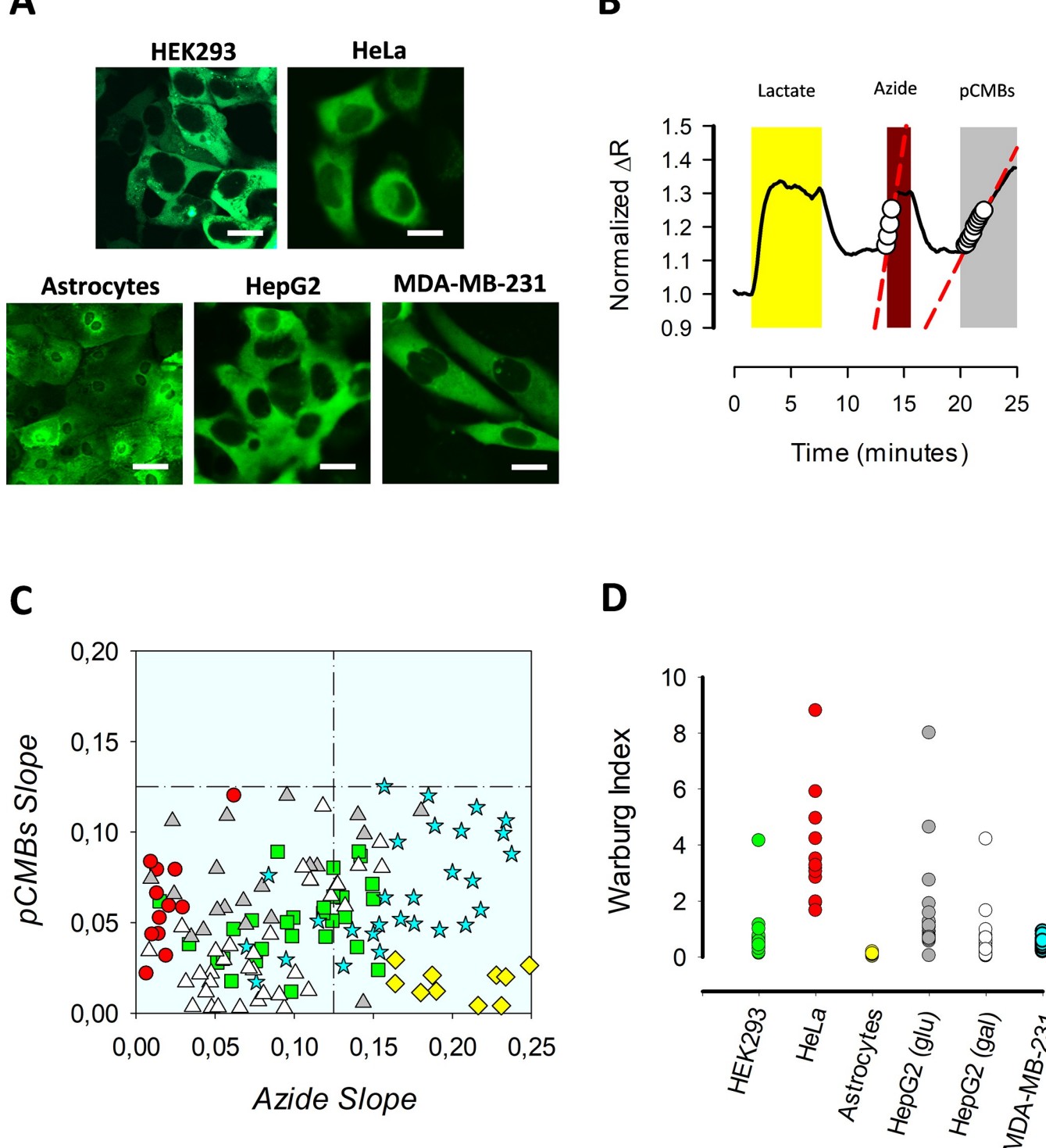

**Fig 2. Selection of an oxidative cell line.** A) A panel of epithelial, tumoral cellular lines, and primary astrocytes expressing cytosolic Laconic. Images were taken by confocal microscopy. Scale barr 10 μm. B) Standard protocol for WI determination. Oxidative and glycolytic metabolism were explored in a primary culture of astrocytes by ETC inhibition with 5 mM Azide and stopping lactate transport with 250 μM, respectively. C) Response distribution of pCMBS slope (glycolysis) and azide (OXPHOS). Oxidative cells are in the right lower quadrant. D) WI calculations for each cell using the quotient from pCMBS/azide slopes. Data from three independent experiments.

accumulation to inhibit lactate exit through the monocarboxylate transporter (MCT). During the first seconds the azide slope is a readout of pyruvate consumption and therefore mitochondrial metabolism and pCMBS promotes lactate accumulation, which is proportional to its production through glycolysis (**Fig 2B**). Within the panel of cell lines, we tested: epithelial HEK293 cells, tumoral cell line HeLa, primary astrocytes from mouse cortex, hepatic cell line HepG2, and tumoral cell line MDA-MB-231. Cells were cultured at 100% confluency with their recommended media. MDA-MB-231 cells were grown using Leibowitz media with galactose as the only carbon source to boost oxidative metabolism.

Cell distribution based on their pCMBS/Azide slopes indicates that MDA-MB-231 are as oxidative as astrocytes, since the major part of both cell types are located in the right lower quadrant (**Fig 2C**), which is an oxidative zone characterized by higher azide slopes than pCMBS. In agreement with previous results, primary culture of astrocytes present WI below one [29], since the WI is the glycolytic/oxidative metabolism quotient, this value confirms that astrocytes are highly oxidative. Using WI from astrocytes as a reference, we determined this parameter for tumoral and epithelia cell lines to find an oxidative system to substitute primary culture of astrocytes. HEK293 cells showed an intermediate WI value (**Fig 2D**), which is in agreement with its epithelial origin and is consistent with previous results [29]. HeLa cells presented a high WI value, consistent with their aggressive nature (**Fig 2D**). The lowest WI values were observed in MDA-MB-231 breast adenocarcinoma and HepG2 hepatocellular carcinoma, cultured in galactose to make them more oxidative (**Fig 2D**), but MDA-MB-231 cells showed a more dramatic increase in intracellular lactate in response to azide (**Fig 2C**). For this reason, we chose MDA-MB-231 cells to generate a report cell line.

## Generation of MitoTox reporter cell

To generate a cell system with a high signal-to-noise ratio suitable for standard multiplate readers, we developed a clonal cell line with high levels and homogeneous Laconic expression. To do so, we produced a lentivirus particle to infect and transduce Laconic in MDA-MB-231 cells. From pooled cells with diverse levels of Laconic expression we selected high expression clones through FACS. Within the high expression clones, we selected over-expressing Laconic clonal MDA-MB-231 cells that showed a homogenous cytosolic and nuclear exclusion expression signal (**Fig 3A**). We termed this cell line "MitoTox Reporter". Exposure of these cells to classical mitochondrial poisons produced a robust increase in the fluorescent signal, indicating

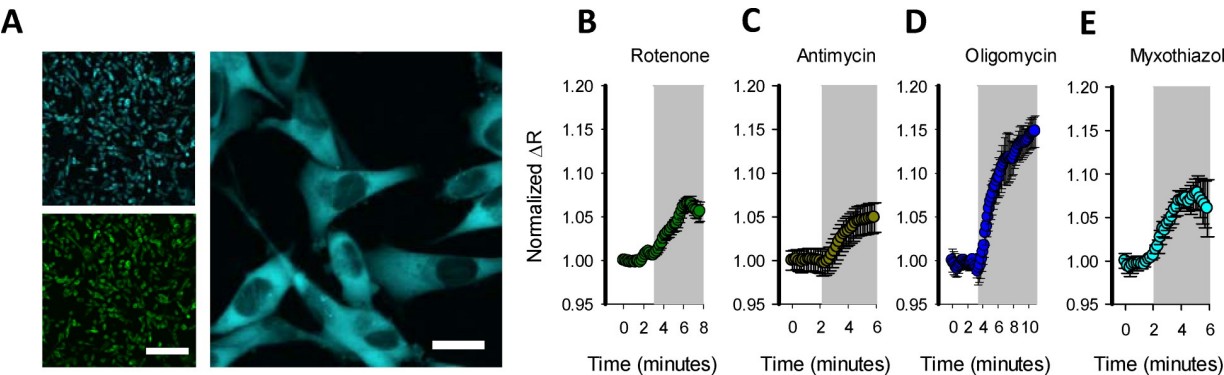

**Fig 3. Characterization of "MitoTox reporter" cell line.** A) Confocal images from FACS selected clonal MDA-MB-231 cells expressing Laconic. Size bars for 20x and 60x represent 100 μm and 10 μm, respectively. Cytosolic lactate accumulation induced by treatment with: B) 32 μM rotenone, C) 16 μM of antimycin, D) 80 μM of oligomycin, and 10 μM myxothiazol. Each trace is the average of ten cells from one representative experiment.

acute toxicity-induced lactate accumulation. For instance, treatment with rotenone, antimycin, oligomycin, and myxothiazol produced an acute, fast, and robust increase of intracellular lactate levels (**Fig 3B, 3C, 3D and 3E**). The response was higher in comparison with the increase of lactate levels induced by mitochondrial toxicants in astrocytes, demonstrating high sensitivity to mitochondrial toxicants. With this evidence in hand, cells were tested in a multi-well plate reader.

## Mitochondrial toxicants produced toxicity-induced lactate accumulation detectable by a multi-plate reader

Fluorescence-based readouts are amenable to HTS applications. To assess the detection of toxicity-induced intracellular lactate accumulation by mitochondrial toxicants in a multiwell plate format, MitoTox reporter cells were seeded in 96-well plates and incubated without $CO_2$ equilibration at 37˚C until they reached a maximal confluency, using a galactose rich Leibovitz medium. Cells were exposed to a high concentration of classical mitochondrial toxicants: 32 µM rotenone, 16 µM antimycin, 5 mM azide, 80 µM oligomycin, and 10 µM of myxothiazol. The effects were measured at 5, 10, 30, and 60 minutes. All these mitochondrial toxicants produced an acute lactate accumulation that was detectable in just 5 minutes at 37˚C. The amplitude of the response was highly variable and based on ΔR% not all the mitochondrial toxicants produced enough lactate accumulation to saturate the sensor, not making it possible to take full advantage of the sensor´s dynamic range (**Fig 4A, 4B, 4C, 4D, 4E and 4F**). Also, the response was not stable over-time especially with azide which declined after 5 minutes (**Fig 4D**). Positive control 10 mM lactate also does not induce enough lactate accumulation to saturate the sensor (**Fig 4A**). However, some wells with higher responses were consistent with the maximal FRET change of Laconic, which is 40%. Each experimental series included a well with the solvents used to prepare the mitochondrial toxicants in order to test if they produced interference. DMSO, ethanol, and KRH buffer did not produce any interference at the concentration used to perform the toxicity experiments, since they did not induce a significant effect on ΔR% over-time (**Fig 4A, 4B, 4C, 4D, 4E and 4F**). Data dispersion was high and ΔR% were not high enough to produce a method suitable for HTS application but gives a proof of the concept that is possible to detect increments of lactate levels induced by a mitochondrial toxicant in a single-well format using a standard multiplate reader. These results indicate that MitoTox reporter cells allow the detection of mitochondrial dysfunction monitoring the increase of ΔR% in a single-well scale, but the response needs to be improved to make the method suitable for HTS.

## Optimization of toxicity-induced lactate accumulation blocking the monocarboxylate transporters

To develop a suitable method for HTS application, it was necessary to increase de amplitude of toxicity-induced lactate accumulation and decrease data variation. In order to tackle these challenges, we boosted the response induced by mitochondrial dysfunction, blocking lactate exit and forcing sensor saturation by pharmacological means. To block lactate efflux, we used pCMBS an inhibitor of MCTs [37], which is a non-permeable sulfhydryl group modifier agent with broad specificity including MCT4, which is the main monocarboxylate transporter present in MDA-MB-231 cells [38]. We added 50 µM pCMBS to block lactate exit at the same time as mitochondrial toxicants such as rotenone, antimycin, azide, oligomycin, and myxothiazol. The effects of each toxicant were measured at 5, 10, 30, and 60 minutes (**Fig 5A, 5B, 5C, 5D, 5E and 5F**). These mitochondrial toxicants produced acute intracellular lactate accumulation in MDA-MB-231 cells tested in a 96 well plate format. The amplitude of the signal with

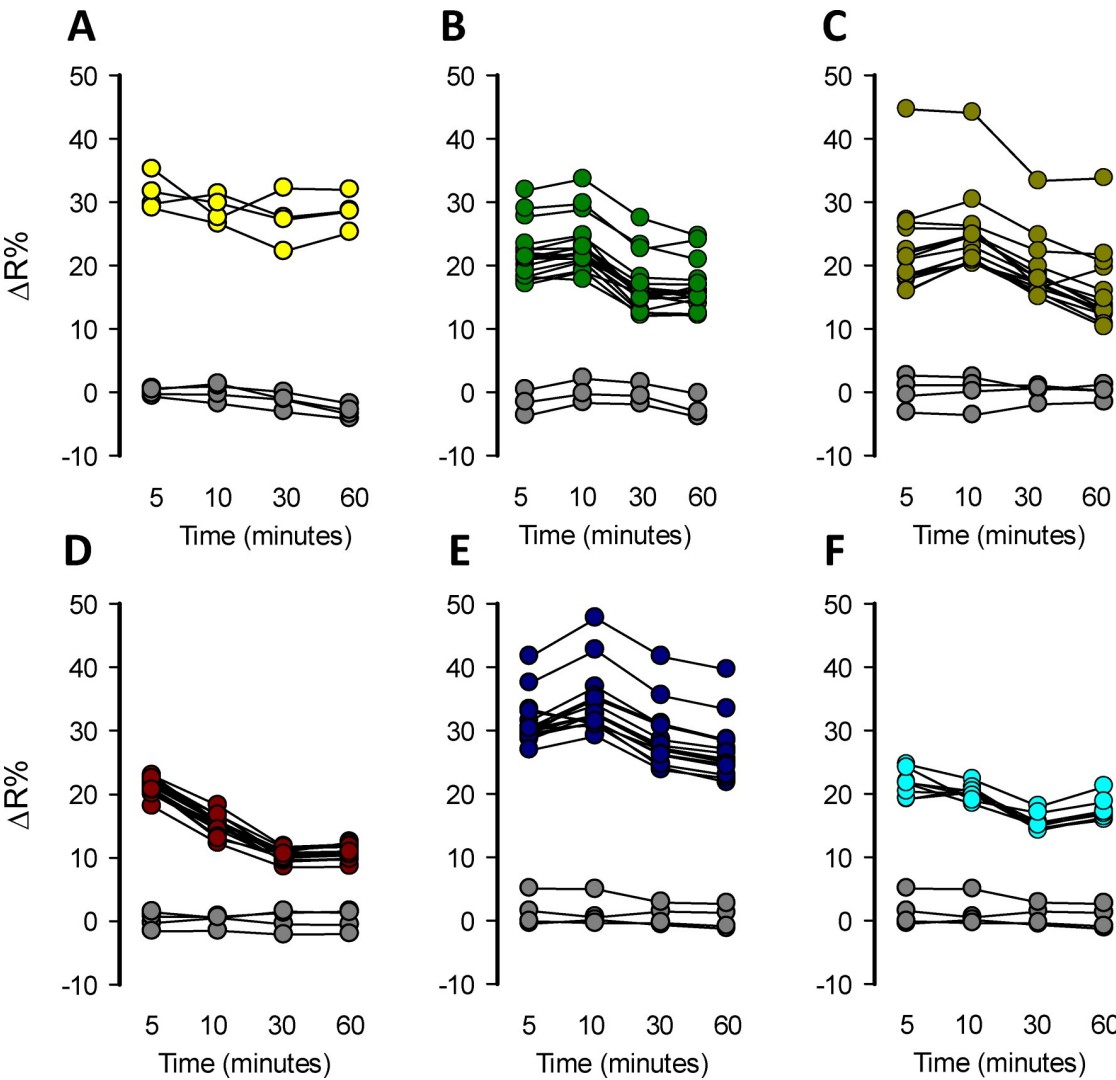

**Fig 4. Single-well detection of lactate accumulation induced by mitochondrial toxicants.** Mitochondrial dysfunction induced by mitochondrial toxicants was detected in MitoTox Reporter cells in 96 well plates. Measurements using a standard multiplate reader were performed at 5, 10, 30, and 60 minutes. A) 10 mM Lactate, B) 32 μM rotenone, C) 16 μM antimycin, D) 5 mM azide, E) 80 μM oligomycin and F) 10 μM myxothiazol. Solvent control (gray circles): KRH buffer for azide, 0.8% DMSO for rotenone, oligomycin, and myxothiazol and 0.16% ethanol for antimycin. All the experiments were performed at 37˚C.

pCMBS was higher and the data present low intra assay variability compared to the experiments without pCMBS. MCT blockage did not produce lactate accumulation, discarding possible interference with basal lactate production. This protocol provided a more robust assay with low variability and higher changes of ΔR% induced for mitochondrial toxicants in a multiplate reader, which are critical parameters to evaluate the suitability of the methodology for HTS applications. The assay was stable at least for 60 minutes for rotenone, antimycin, and myxothiazol (**Fig 5B, 5C and 5F**), 30 minutes for azide (**Fig 5D**) and 10 minutes for oligomycin (**Fig 5E**).

To evaluate the suitability of the improved assay for HTS applications, a coefficient called the Z'-Factor was calculated [32]. This coefficient reflects the assay´s signal dynamic range and the data variation associated with the signal measurements. Z'-factor values between 0.5–1

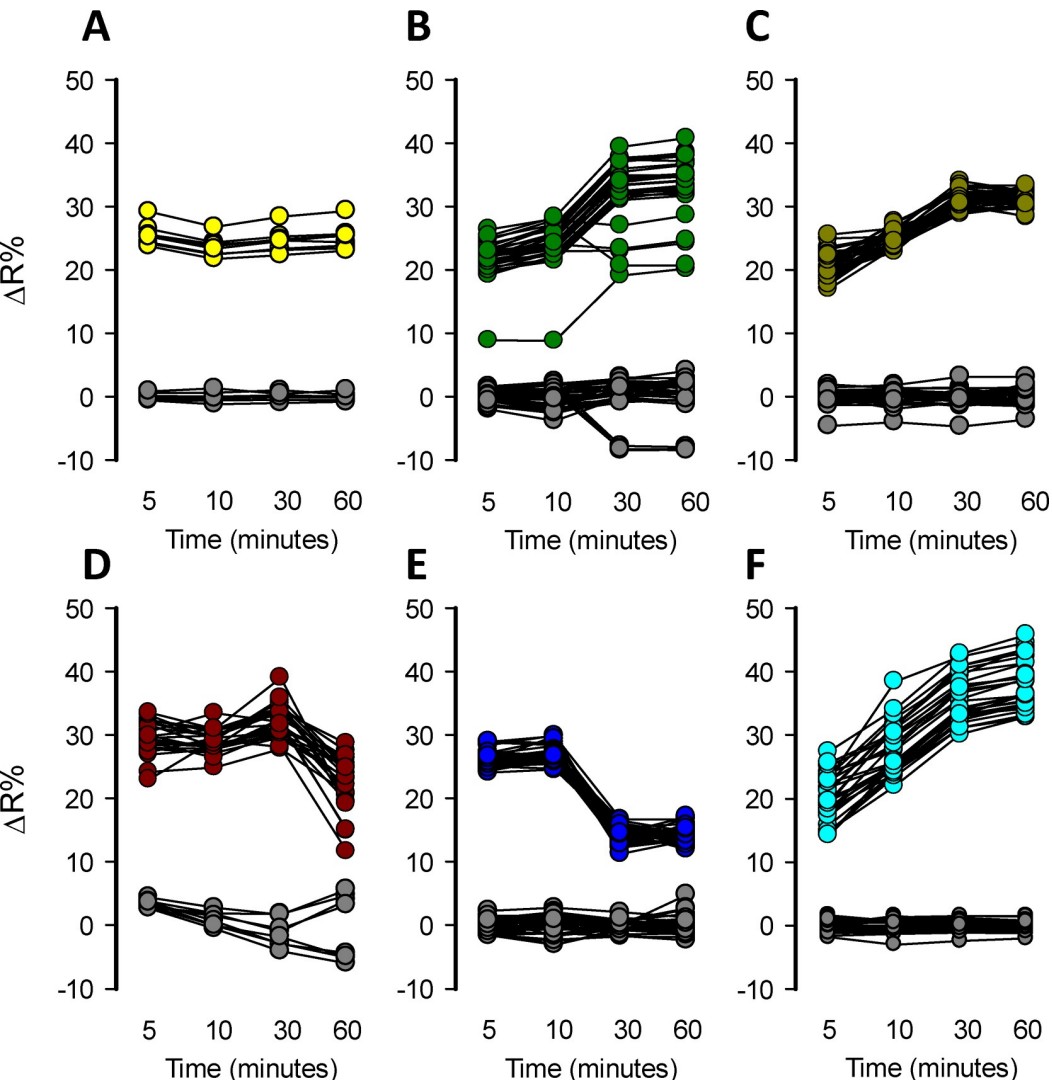

**Fig 5. Enhancement of lactate accumulation by transport-stop protocol.** All the experiments were performed adding 50 μM of pCMBS together with the mitochondrial toxicant. Measurements were performed at 5, 10, 30, and 60 minutes. A) pCMBS control, B) 32 μM rotenone, C) 16 μM antimycin, D) 5 mM azide, E) 80 μM oligomycin and F) 10 μM myxothiazol. Solvent control (gray circles): KRH buffer for azide, 0.8% DMSO for rotenone, oligomycin, and myxothiazol and 0.16% ethanol for antimycin. All the experiments were performed at 37°C.

means that a hit can be identified with a confidence of 99.73–100% using a single-well. Values below 0.5 indicate the necessity to perform duplicates or triplicates for each molecule and values below 0 indicated the unsuitability of the assay for HTS application. We performed a series of experiments to calculate the Z'-factor using different mitochondrial poisons and replicate plates on different experimental days. Treatment with each mitochondrial toxicant produced acute lactate accumulation with low intra-well variability and a high ΔR% increase almost reaching the reporter maximal fluorescent change. All our assays using different mitochondrial toxicants reached Z'-factor above 0.3. Majority of our replicates reached a Z-Factor over 0,5 supporting the suitability of the assay to identify a mitochondrial toxicant in a single-well screening (**Fig 6**). For all mitochondrial toxicants tested the better Z'-Factor was obtained 10 minutes after treatment. The intra-plate variability for the improved method was determined

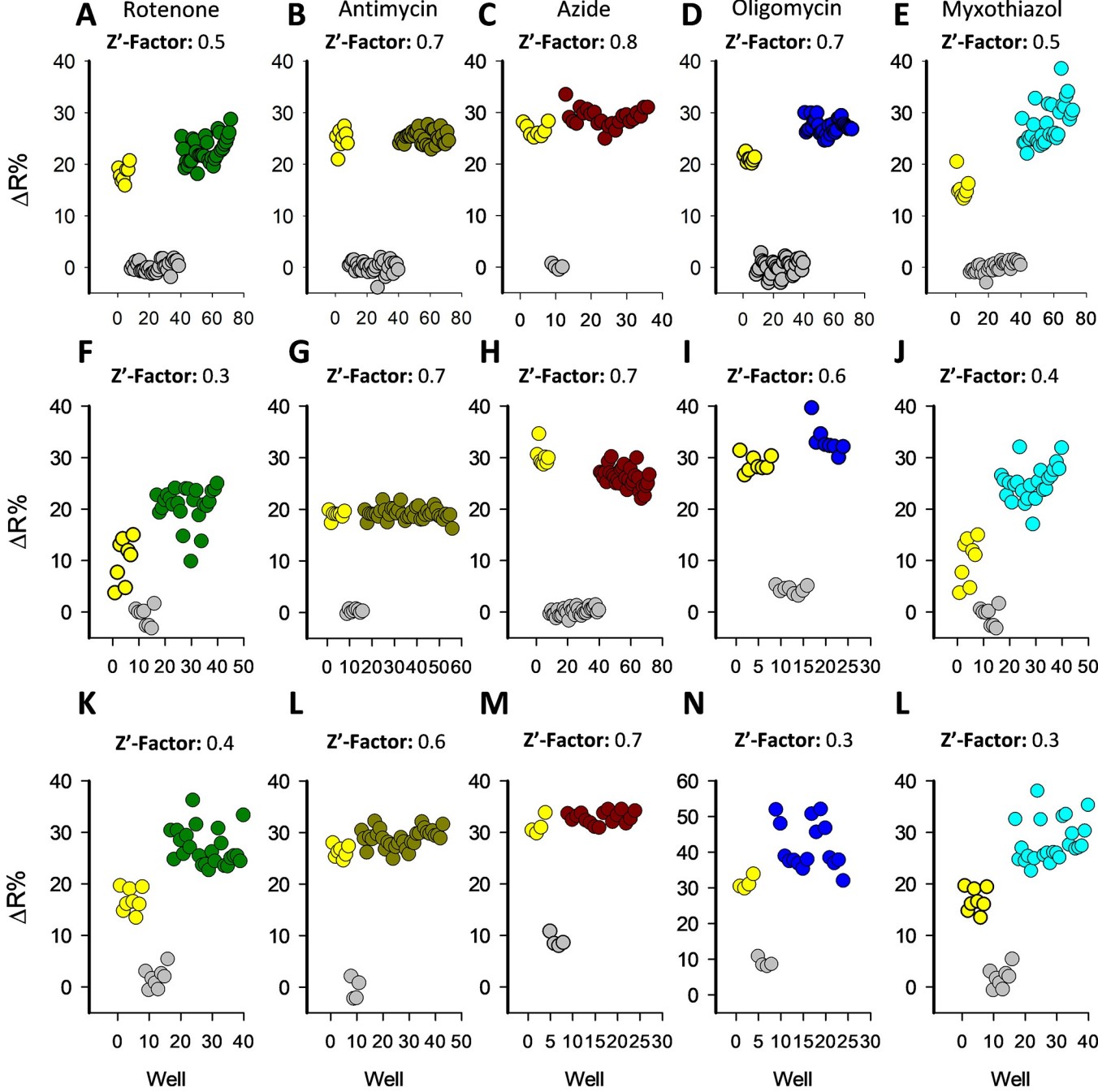

**Fig 6. High throughput suitability characterization.** The Z'-factor was calculated using wells treated with 32 μM rotenone, 16 μM antimycin, 5 mM azide, 80 μM oligomycin, and 10 μM myxothiazol as positive controls. Negative control wells were treated with 50 μM of pCMBS (gray circles) and nanosensor response control with 10 mM lactate (yellow circles). All the Z'-Factors were calculated at 10 minutes post-treatment. Experiments from three independent assays for each mitochondrial toxicant, performed on different experimental days.

using the % of Variation Coefficient (CV). All the mitochondrial toxicants and positive control had an intra-plate CV% below 20 and comparison between CV% values from different plates indicate low inter-plate variability (Fig 7). Our date shows that the improved method using a pharmacologic approach to block the lactate exit is suitable for HTS applications.

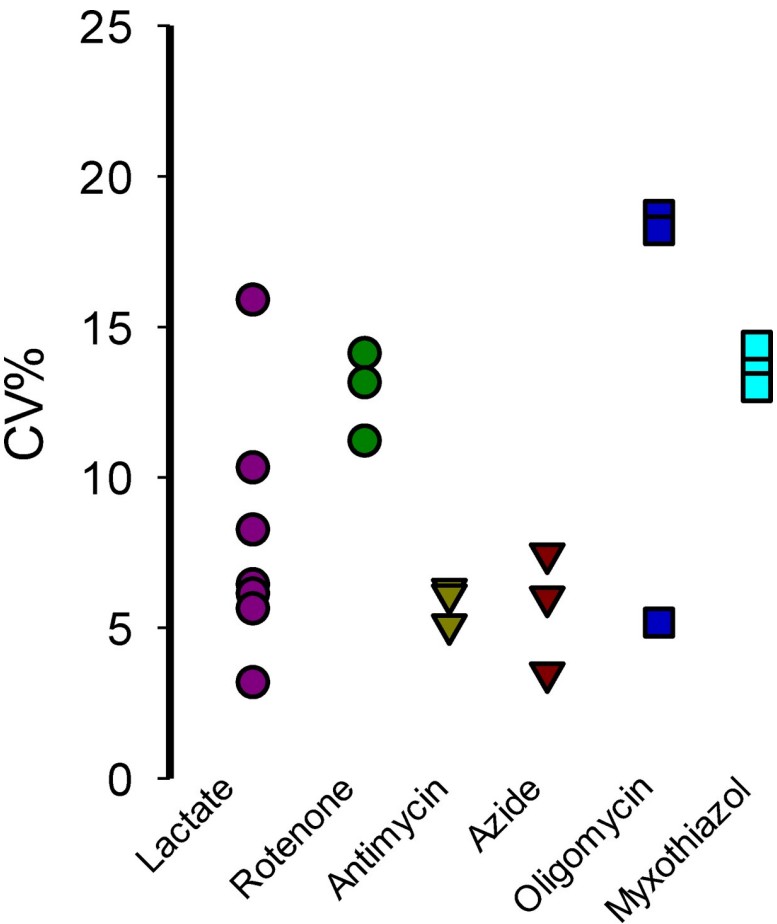

**Fig 7. Inter and intra-well variability analysis.** CV% values were calculated using the average wells from three independent plates.

## Sensitivity of mitochondrial toxicity detection

To assess the sensitivity of our methodology we determined the $IC_{50}$ for a panel of classical mitochondrial toxicants. Dose-response curves were constructed by treating the cells with an increasing concentration of rotenone, antimycin, azide, oligomycin, and myxothiazol. We measured the toxicity-induced lactate accumulation at 30 minutes of treatment. To quantitatively evaluate the potency of these compounds, we fitted our data to a rectangular hyperbola to calculate the $IC_{50}$ for each toxicant. We determine that the complex III inhibitor antimycin was the most potent mitochondrial toxicant from our panel with an $IC_{50}$ of 0.24 ± 0.48 nM (**Fig 8B**), followed by complex I inhibitor rotenone which has an $IC_{50}$ 2.7 ± 1.41 nM (**Fig 8A**). Complex III myxothiazol reached an $IC_{50}$ of 1.2 ± 0.55 μM (**Fig 8E**) and the ATPase inhibitor oligomycin reached $IC_{50}$ of 38 ± 15 nM (**Fig 8D**). The weakest molecule of our mitochondrial toxicants panel was azide, a cytochrome c inhibitor which reached an $IC_{50}$ of 1.44 ± 0.76 mM (**Fig 8B**). To gain a wider view about the sensitivity of our method, we confronted the $IC_{50}$ from our methodology with the state-of-the-art methods to assess mitochondrial dysfunction (**Table 1**). Our methodology allows the detection of mitochondrial toxicity induced by classical toxicants faster than ATP, MMP, GSH, viability and ROS measurements and present improved sensitivity compared to OCR and ECAR when detecting mitochondrial dysfunction induced by rotenone, antimycin, and oligomycin using seahorse technology. These results

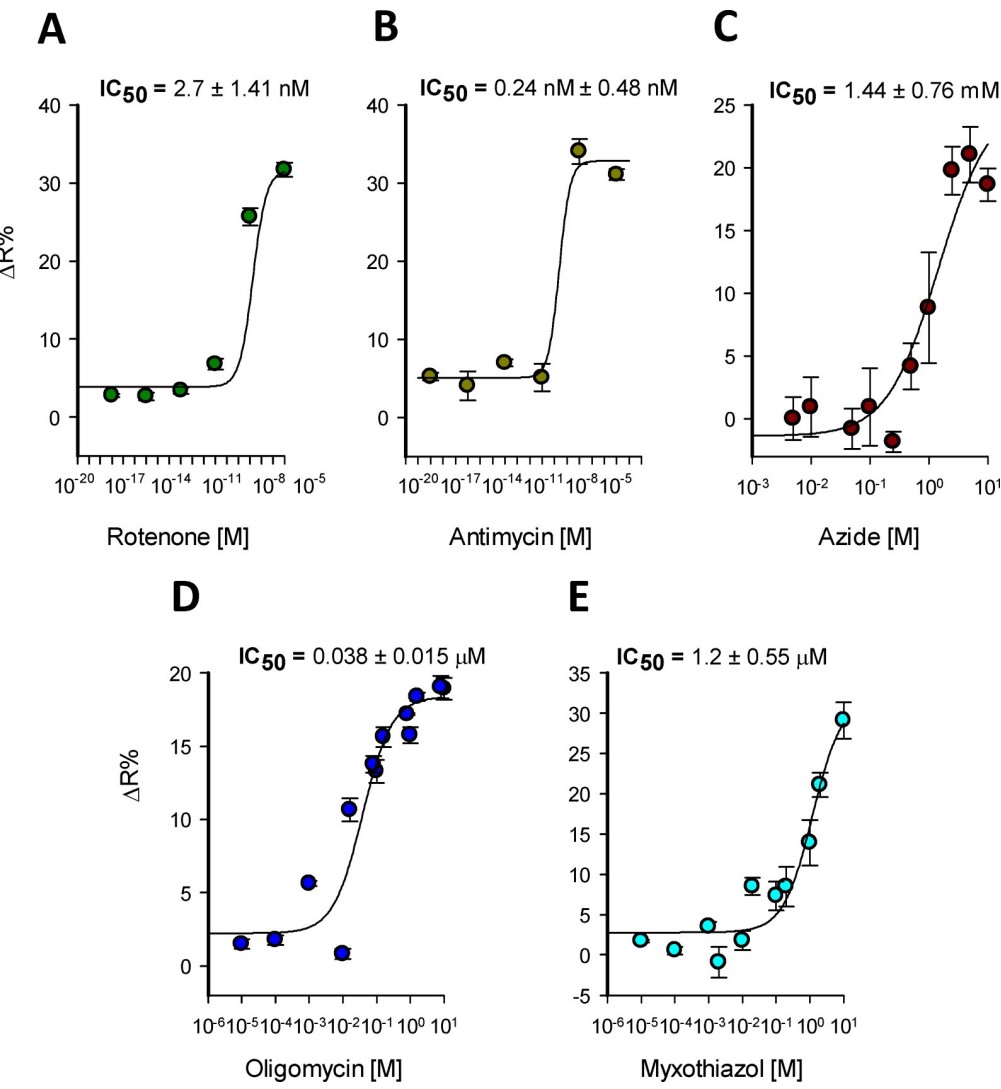

**Fig 8. IC$_{50}$ for classical mitochondrial toxicants.** Dose-response Lactate accumulation induced by classical mitochondrial toxicants was measured at: A) $10xE10^{-5}$, $10xE10^{-8}$, $10xE10^{-11}$, $10xE10^{-14}$, $10xE10^{-17}$ and $10xE10^{-20}$ in molar of rotenone and B) antimycin C) 10, 5, 2.5, 1, 0.5, 0.25, 0.1, 0.05, 0.01 and 0.005 in mM of azide D) 16, 10, 8, oligomycin and E) 10, 2, 1, 0.2, 0.1, 0.02, 0.01, 0.002, 0.001, 0.0001 and 0.00001 in μM of myxothiazol. Data was collected at 10 minutes after toxicant treatment at 37°C. Data is the average from three independent experiments.

suggest that real-time monitoring of intracellular lactate levels is more sensitive and faster than the state-of-the-art technology to evaluate mitochondrial dysfunction using standard multiplate readers.

## Pilot screening

The capability of the assay to identify previously described toxic drugs was explored through a pilot screening assay. We selected 13 compounds from a variety of therapeutic classes to perform a pilot screening assay using MitoTox Reporter cells and a standard multiplate reader. Among the selected compounds we included thiazolidinediones such as ciglitazone, troglitazone, and rosiglitazone which are antidiabetic agents with known organ toxicity [39], camptothecin a topoisomerase inhibitor that produces hepatic toxicity [40], prazosin an alpha-

**Table 1. IC$_{50}$ Comparison of state-of-art methods to evaluate mitochondrial dysfunction.** Benchmarking analysis comparing the IC$_{50}$ archived with the Laconic based method and current technology to assess mitochondrial dysfunction.

| REFERENCE | METHOD | TREATMENT TIME (hours) | Rotenone IC50 (μM) | Antimycin IC50 (μM) | Azide IC50 (mM) | Oligomycin IC50 (μM) | Myxothiazol IC50 (μM) |
|---|---|---|---|---|---|---|---|
| This Study | MitoToxy | 0.5 | 0.0025 | 0.00024 | 1.44 | 0.038 | 1.2 |
| (Nadanaciva et al., 2012)[55] | ECAR | 0.03 | 0.371 | 0.1 | ND | 1.1 | ND |
| | OCR | 0.03 | 0.263 | 0.004 | ND | 0.92 | ND |
| (Marroquin et al., 2007)[35] | ATP* | 24 | 0.05 | 0.006 | ND | 0.005 | ND |
| (Li et al., 2014)[50] | MMP | 6 | 20.3 | 2.1 | ND | 15.6 | 1.2 |
| | GSH | 6 | 518.3 | 227.8 | ND | 160.9 | >1000 |
| | Viability | 6 | >1000 | 290.6 | ND | 181.8 | 183.4 |
| | ROS | 6 | 10.5 | 17.7 | ND | >1000 | 3.3 |
| (Hynes et al., 2013)[72] | Oxygen | 0.13 | 0.020 | 0.039 | ND | 0.038 | ND |
| | ECA | 0.13 | 0.002 | 0.074 | ND | 0.077 | ND |

*Approximated IC50 directly from plotted data in the original paper. ND: No Determined

adrenergic antagonist with no reported associated toxicity, simvastatin a long-established hydroxy-methylglutaryl coenzyme A (HMG-CoA) reductase inhibitor safe if administered alone [41], metformin a safe antidiabetic agent [42], terfenadine a worldwide withdrawn anti-histamine due to cardiac arrhythmia [43], a nonsteroidal antiandrogens nilutamide, and fluta-mide which are used for the treatment of prostate cancer and produced a decrease of oxygen consumption and hepatotoxicity [44, 45], etoposide semisynthetic derivative of podophyllo-toxin with broadly anticancer activity with low side effects [46], acetylsalicylic acid and lido-caine both with no reported cell toxicity effects. To perform the assays, we prepared master stocks for each compound in 0.2% DMSO. Assays were performed with compounds at final concentrations of 1, 5, and 10 μM and the response of MitoTox Reporter cells were measured at 5, 10, 30, and 60 minutes. Thiazolidinediones such as ciglitazone, troglitazone, and rosiglita-zone produces a robust increase over 14% of ΔR at 10 μM, which is consistent with an acute increase of cytosolic lactate due to mitochondrial dysfunction. The effect was also detected at 5 μM for troglitazone (**Fig 9A and 9B**). Lactate accumulation was stable during 60 minutes after drug treatment at 10 μM of each compound (**Fig 10**). These results are consistent with the known cellular toxicity identified for these class of molecules [47]. Additionally, worldwide withdrawn antihistamine terfenadine produced a rapid and robust increase of 15.64 ± 1.32 of ΔR% at 10 μM, without any measurable effects at 1 or 5 μM (**Fig 9A and 9B**). Lactate accumu-lation was stable during 60 minutes after drug treatment at 10 μM (**Fig 10**). Strikingly, a highly toxic anti-cancer drug camptothecin produced a potent increase of 49.18 ± 7.4 ΔR% at 10 μM consistent with high lactate accumulation into the cytosol (**Fig 9A and 9B**). Taking into account the reported dynamic range and sensitivity of laconic [29], camptothecin produced lactate accumulation over 10 mM since the drug treatment induced a maximal change in the dynamic range of the sensor. Also, a potent effect at 5 μM was observed, but no change in ΔR% was detected at 1 μM of the drug. The effects on lactate metabolism were stable during 60 minutes after drug treatment at 10 μM (**Fig 10**). Also, nilutamide and flutamine with known hepatotoxic effects, showed an opposite effect on lactate metabolism. Although nilutamine did not produce any effect, flutamine induced an 8.71 ± 2.06 ΔR% at 10 μM. As a negative control we selected a compound with no reported side-effects. Among the selected molecules we had metformin, prazosin, simvastatin, etoposide, aspirin, and lidocaine. None of these molecules produced an important ΔR% variation and therefore lactate accumulation at 1, 5, and 10 μM (**Fig 9A and 9B**). These results validate our methods to identify molecules with a toxic profile.

## A

| 10 minutes | | (ΔRmax-ΔRmin)/ΔRmin*100 | | | | | | | | | |
|---|---|---|---|---|---|---|---|---|---|---|---|
| | 2 | 3 | 4 | 5 | 6 | 7 | 8 | 9 | 10 | 11 | 12 |
| A | 2,2 | 0,4 | -2,2 | 10,4 | 17,4 | -2,2 | 2,1 | 0,6 | -0,1 | 7,9 | 15,1 |
| B | 0,8 | 1,8 | -0,8 | 9,5 | 18,7 | -2,9 | 3,1 | 2,2 | -1,5 | 5,8 | 16,2 |
| C | 1,4 | 0,5 | -1,6 | 9,4 | 18,9 | -2,0 | 4,3 | 2,4 | -0,9 | 5,6 | 13,7 |
| D | 0,2 | 1,5 | -0,1 | 8,4 | 15,9 | -3,7 | 4,0 | 3,0 | -2,1 | 3,1 | 13,9 |
| E | -0,6 | 0,5 | -2,6 | 2,6 | 4,8 | 3,9 | 13,0 | 12,6 | -4,1 | -2,0 | 14,6 |
| F | -0,3 | 1,3 | -2,5 | 3,5 | 4,0 | 0,3 | 13,9 | 13,3 | -2,5 | -0,5 | 14,6 |
| G | 0,6 | -0,1 | -2,8 | 2,7 | 4,2 | 5,2 | 15,6 | 15,7 | -2,5 | -0,3 | 16,0 |
| H | 0,3 | 2,0 | -0,3 | 3,3 | 4,5 | 3,6 | 17,8 | 17,6 | -1,7 | 2,2 | 17,4 |
| | Prazosin | | Ciglitazone | | | Metformine | | | Rosiglitazone | | |
| | Ctrl | | Sinvastatine | | | Troglitazone | | | Terferadine | | |

| 10 minutes | | (ΔRmax-ΔRmin)/ΔRmin*100 | | | | | | | | | |
|---|---|---|---|---|---|---|---|---|---|---|---|
| | 2 | 3 | 4 | 5 | 6 | 7 | 8 | 9 | 10 | 11 | 12 |
| A | -1,6 | -2,1 | 18,3 | 58,3 | -5,5 | 2,0 | -2,6 | 0,3 | 1,4 | 6,0 | 1,1 |
| B | -1,1 | -0,9 | 14,3 | 52,1 | -3,6 | 0,8 | -0,3 | 0,1 | 1,8 | 9,2 | -0,2 |
| C | -0,2 | -0,6 | 11,3 | 43,5 | -4,0 | 1,6 | 1,2 | 1,5 | 3,5 | 11,0 | 0,8 |
| D | -0,8 | -1,4 | 10,2 | 42,9 | -4,2 | 2,0 | -0,9 | 1,9 | 3,5 | 8,6 | -0,4 |
| E | -0,2 | 3,2 | 4,5 | 5,4 | 1,1 | 0,7 | 0,8 | -4,0 | 2,8 | 2,9 | 1,3 |
| F | -0,7 | 2,9 | 4,2 | 4,3 | 0,4 | 2,2 | 2,4 | -3,4 | 4,1 | 3,5 | -0,2 |
| G | -1,2 | 0,5 | 3,1 | 4,1 | 2,2 | 2,1 | 0,9 | -4,0 | 3,8 | 4,3 | 0,7 |
| H | 1,0 | 2,3 | 4,7 | 2,1 | 1,6 | 2,2 | 3,0 | -2,5 | 6,3 | 8,4 | 1,8 |
| | Ctrl | Camptothecin | | | Etoposide | | | Flutamide | | | Ctrl |
| | | Nilutamide | | | Aspirin | | | Lidocain | | | |

## B

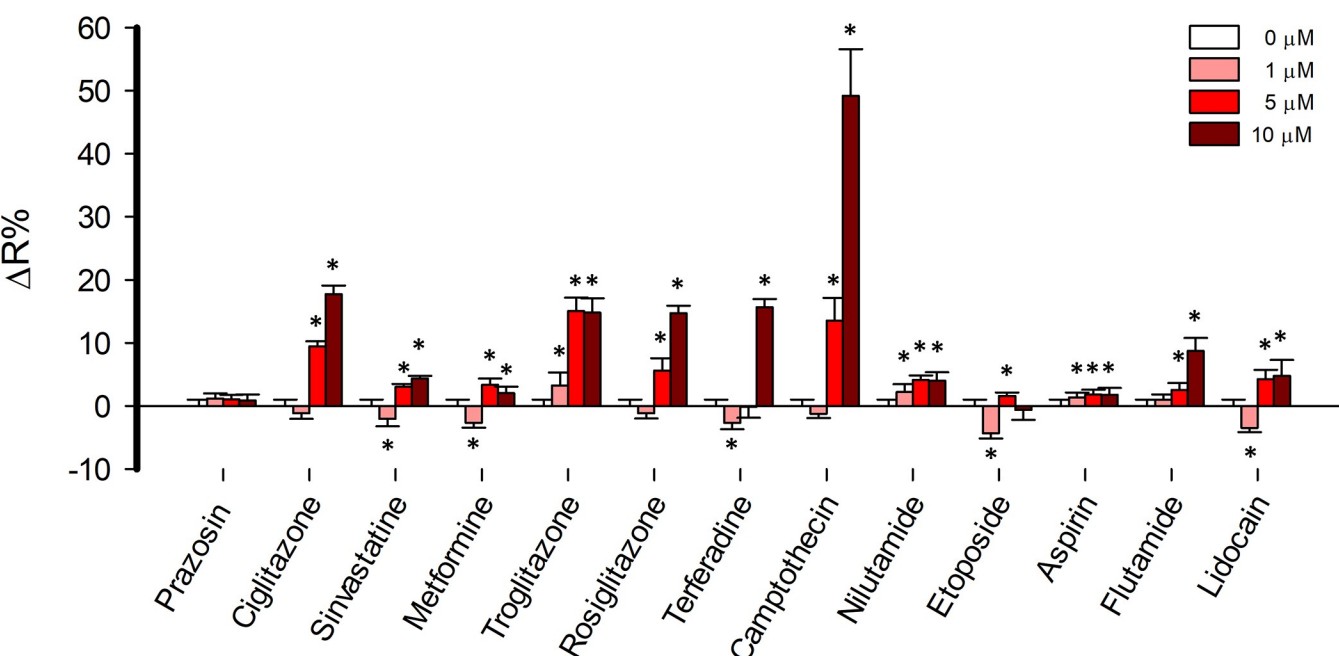

**Fig 9. Pilot screening.** A) Schematic representation of the 96 well plate format with their corresponding ΔR% values in pseudo color codification. Red and blue colors represent high and low ΔR%, respectively. Drugs were used at 1, 5 and 10 μM and effects were measured after 30 minutes of treatment at 37˚C B) Barr plot of the effect on lactate levels of a panel of drugs at 1, 5, and 10 μM at 37˚C after of 30 minutes of pharmacological treatment. Data represent the average of 4 replicate wells of one representative experiment.

## Discussion

This paper describes a phenotypic high-throughput screening method for the identification of mitochondrial toxicants from chemical libraries. The methodology takes advantage of toxicity-induced intracellular lactate accumulation monitored using a genetically encoded lactate indicator, Laconic. To test the methodology, lactate accumulation was induced by acute pharmacological blockage of ETC at different levels by classical mitochondrial toxicants. The suitability for HTS applications was tested and sensitivity was evaluated with the current state-of-the-art approaches to assess mitochondrial dysfunction. Pilot screening was performed to evaluate the capabilities of the methodology to identify previously described toxic drugs.

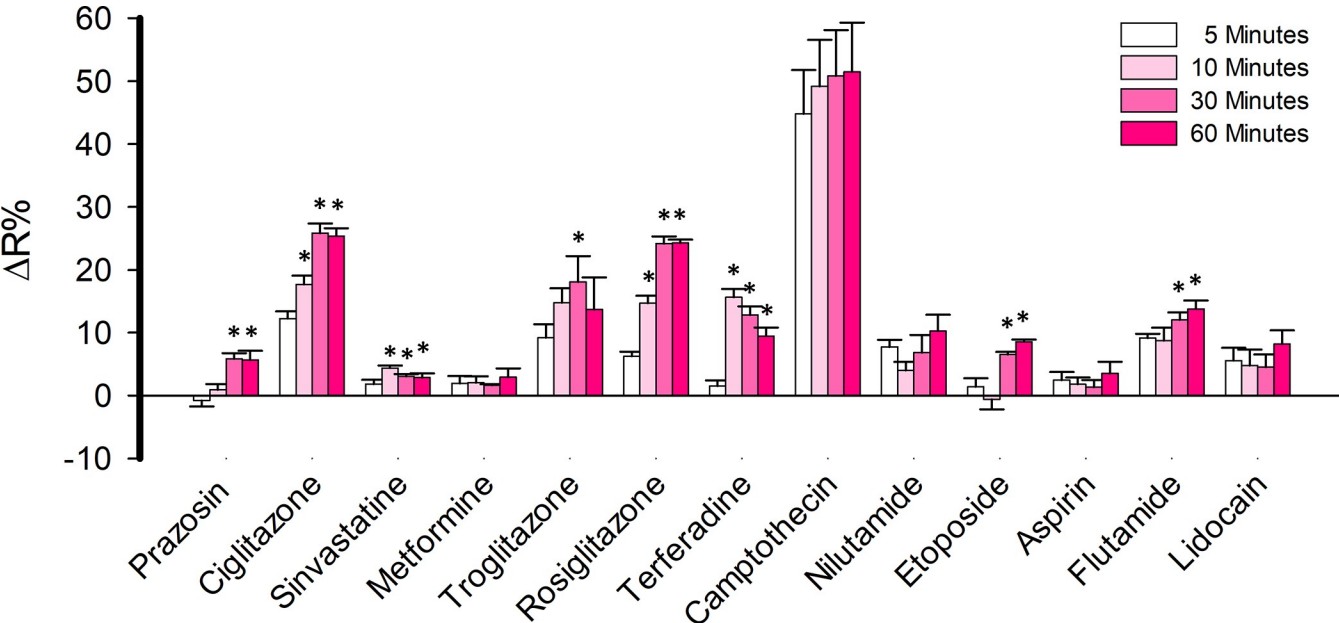

**Fig 10. Stability of lactate accumulation.** Lactate accumulation at 5, 10, 30, and 60-minutes of treatment with a panel of drugs at 37 C. Data obtained from treatment with 10 μM of each compound. Data represent the average of 4 replicate wells of one representative experiment.

Despite the efforts to develop and optimize a high-throughput method to assess mitochondrial dysfunction, there is still space for improvements. For instance, Seahorse® the most commonly used technology to evaluate oxidative and glycolytic metabolism through oxygen consumption and extracellular acidification measurements, has limitations: i) it requires an expensive dedicated equipment and sophisticated consumables, which increases the cost of the assays, ii) it is not possible to scale up to a 384 and 1536 well format, increasing the cost and time for large screening campaigns and iii) assays must be performed without bicarbonate, which is a physiological buffer system, critical for metabolic regulation in mammalian cells [48]. Additionally, methods to evaluate MMP using TMRM, ATP using bioluminescence, ROS production with Mito-SOX, cell viability using calcein, and GSH production with Monobromobimane (mBBr) can be scaled up to the 384 well format but display low sensitivity. For instance, MMP and ROS are not always coupled to ATP production due to the proton leak [49]. Phenomena that is explained as protons return to the mitochondrial matrix independently of ATP synthase, increasing the probability of false-negative results. Therefore, these assays cannot distinguish whether such a loss is due to inhibition, uncoupling, or MPT (mitochondrial permeability transition) [9]. Furthermore, it has been shown that ROS, GSH, and viability assays have low sensitivity compared to MMP measurements to identify mitochondrial toxicants such as thiazolidinediones and camptothecin [50].

Development of a sensitive cell-based assay to assess mitochondrial dysfunction for HTS uses requires the selection of a highly oxidative and low lactate permeability cell system that promotes intra-cellular lactate accumulation. We took advantage of our previously described WI [29] to evaluate a panel of cell lines and media culture conditions to identify oxidative cell lines. Our results are consistent with the WI obtained previously and show that this parameter is a good alternative methodology for evaluating glycolytic and oxidative metabolism with a high spatial-temporal resolution. Indeed, the oxidative nature of MDA-MB-231 cell growth in galactose rich media detected by WI is consistent with previously published results using Seahorse® technology [51].

Different organs have different susceptibilities to mitochondrial toxicants. Mitochondrial damage typically affects highly oxidative tissues, such as kidney and heart, or tissues exposed to higher drug concentrations, such as the liver [9]. Hepatotoxicity is a significant cause of Drug-Induced Liver Injury (DILI) [52], being responsible for 11% of pre-clinical failures [53], 33% of clinical toxicity [53], and 18% of post-marketing withdrawal [7]. Our analysis of glyco-lytic and oxidative metabolism shows that the standard cell type used for toxicity assays, liver carcinoma HepG2, presents a small response to mitochondrial inhibition, which is not ideal for the present protocol. However, HepG2 cells did show a low WI, a parameter that is more cumbersome to estimate but could be applied for the detection of mitochondrial toxicity. Interestingly, it has been reported that likewise HepG2 cells MDA-MB-231 cells express Phase 1 and Phase 2 metabolizing enzymes [54].

Despite the fact that our methodology is based on a breast cancer cell line, our results from $IC_{50}$ determination using classical mitochondrial toxicants show that is more sensitive and requires less time for readout measurements. Indeed, our method is capable of detecting mito-chondrial dysfunction using nanomolar concentrations of the toxicant rotenone in just 30 minutes, much better than current techniques that require hours of pre-treatments. This improved sensitivity suggests that our methodology could be useful for detecting subtle changes in mitochondrial physiology, which are responsible for the appearance of side-effects well after drug approval during the commercialization phase. Additionally, our pilot screening showed the capability of the methodology to detected previously described mitochondrial toxi-cants with improved sensitivity. For instance, the antidiabetic agents thiazolidinediones induced a robust 20% ΔR change at 10 µM that covers half of the sensor´s dynamic range. Pre-vious studies using Seahorse technology show that concentrations over 100 µM of ciglitazone and troglitazone are required to change the OCR and ECAR of HepG2 cells by 50% and no effects were detected between 0.13 and 100 µM for rosiglitazone [55]. Measurements of MMP showed that the $IC_{50}$ for ciglitazone, rosiglitazone, and troglitazone are 135.7, >1000 and 175.6 µM, respectively [50]. The hitherto most sensitive detection of thiazolidinedione toxicity was obtained for troglitazone, which induced ATP depletion with an $IC_{50}$ at 28 and 31,4 µM [56, 57]. The antihistamine drug terfenadine induced a 20% ΔR change at 10 µM, meanwhile MMP measurements had a $IC_{50}$ of 4.4 [50]. The highly toxic anti-cancer drug camptothecin produced a potent increase of 49.18 ± 7.4 ΔR% at 10 µM and 13.52 ± 3.6 at 5 µM. Spite of their known cellular toxicity, this robust response in intra-cellular lactate accumulation over 10 mM is not obvious, since MMP measurements determine the $IC_{50}$ of 205.6 µM [50]. Our negative control molecules metformin, etoposide, aspirin, and lidocaine, which are molecules without reported cellular toxicity, did not induce intra-cellular lactate accumulation, in agreement of previous results [50]. The exceptions were prazosin and simvastatin, which were not detected by our methods, but they induced MMP depolarization with an $IC_{50}$ of 6.3 and 63.2 µM, respectively [50]. These results can be considered false positive hits from MMP measurements, because they are considered to be relatively safe drugs [58]. Also, nilutamide did not produce ΔR% changes at all the concentrations tested, in spite of its known toxicity profile [44], being a false-negative result. A possible explanation for this behavior is that the drug stops pyruvate consumption, but without secondary glycolytic activation, so the amount of lactate is not enough to produce saturation of the sensor. In this case we expect to improve the sensitivity using a version of Laconic that saturates in the µmolar range. Our pilot screening detected the toxicity of ciglitazone, troglitazone, rosiglitazone and camptothecin at 5 µM, whereas the mito-chondrial effects of terfenadine and flutamide were detected at 10 µM. As a reference, the max-imum concentration of these drugs in plasma (Cmax) lies in the range of 3 to 10 µM [59–63]. Together these results indicate that our methodology is complementary with current methods to evaluate mitochondrial function such as MMP measurements, therefore we consider that

potential hits should be validated with complementary techniques. Although well-known mitochondrial inhibitors elicited a prompt Laconic response (within 5 min), not all toxic compounds might show the response so rapidly. To minimize the frequency of false negatives results, we recommend measurements to be carried out at or after 30 minutes of exposure. The present methodology is well suited for the detection of acute mitochondrial toxicity, toxicity due to longer exposure in the range of days of a given toxicant were not tested in this work.

Mitochondrial toxicity arises due to the confluency of multiple toxic insults that produce a hemostasis misbalance on mitochondrial physiology. It seems to be plausible that convergent assays without a focus on molecular targets will be more efficient at identifying toxic molecules. Despite that the first created drugs came from naturally occurring chemistry and were found through phenotypic screening, breakthroughs in DNA recombinant technology, combinatorial chemistry, and liquid handling systems moved the industry to a reductionist approach focusing on preselected molecular targets or single target paradigm [64, 65]. In the last decade, this approach resulted in a dramatic decline in the development rate of new first-class drugs [65, 66]. Currently, phenotypic and convergent readouts are back in the industry, mainly because the ability and technology to identify target molecules responsible for a given phenotype have been improved by means of reverse engineering [67–69]. In this context our methodology based on a genetically encoded lactate sensor, represents a step forward in the development of a new generation of cell-based phenotypic assays for the pharmaceutical industry. Indeed, drug discovery pipelines using genetically encoded sensors for other molecules have been successfully described [24–26, 28, 70], supporting the potential suitability of these types of analytics tools in HTS applications.

The introduced methodology based on a FRET lactate sensor, presents limitations. We obtained Z'-Factors > 0.5, since the maximal amplitude of fluorescent change that Laconic can afford is 40% and the intra-plate variability, we are within the limit of an appropriate Z' Factor for HTS applications. Therefore, if the data variability undergoes a slight increase this will produce a major impact on the Z' Factor, making it difficult to obtain values > 0.5, hampering the suitability of the methodology for HTS applications. To achieve a high Z' Factor in single-well format, it was necessary the prevent the efflux of lactate with pCMBS. However, this compound is not specific for the lactate transporter and might in principle interfere with the import of a given toxicant. Thus, such toxicant would become a false negative. For this reason, in the case of hydrophilic libraries we recommend to run the screening without pCMBS. To preserve sensitivity, the assay would have to be performed in triplicate. Alternatively, MCT4-specific blockers could be used when they become available. Cell-based method required the selection of an over expression clone at enough levels to obtain high signal-to-noise ratio. Specifically, signal-to-noise ratio for mTFP and Venus channels below 1.3 will produce data with high variability. Therefore, establishment of a high expression clonal cell line through FACS is a critical step for method development. Clone selection and implementation can be challenging, since some cell types are not suitable for high protein expression reaching low concentrations of the fluorescent sensor. Additionally, high expression clones are often unstable, limiting the temporal window for the assays to be performed. This can be overcome by inducing an acute and transient transduction of the sensor using another virial delivery system, for instance baculoviral infection. Another limitation is the steady-state fluorescent measurements which are sensitive to color interferences, therefore colored compounds cannot be evaluated using this methodology and they must be discarded previously to avoid false positive/negative hits. Any assay based on cultured cells like the present one risks overlooking toxicants that are tissue specific. Not much is known about inter tissue mitochondrial variability. Mitochondrial toxicity could manifest over extended periods which may not be detected by our assay.

Today, the majority of drug companies detect mitochondrial toxicity measuring oxygen, pH, ATP or MMP in cells cultured in the presence of galactose (glu/gal screen) to make them more oxidative and sensitive to mitochondrial damage [35, 55, 71–73]. The present method is a variation of the glu/gal assay in which the readout is intracellular lactate. The relative advantages of lactate detection in terms of sensitivity, speed of detection and cost have been discussed above. Table 1 shows that the lactate method is as sensitive as the best option in current use, i.e. the oxygen/extracellular acidification method. Parallel assays with these two options will be required to compare them in terms of specificity and selectivity.

Phenotypic and convergent readouts are required to evaluate cellular toxicity at the beginning of the drug development process to decrease the attrition rate in the pharmaceutical industry. Genetically encoded indicators are emerging tools that allow cell physiology in intact systems to be evaluated. We envisage that the continuous development and optimization of sensors for metabolites, focusing on improvements in the dynamic range, brightness, and affinity for HTS purposes, will allow the development of a robust phenotypic cell-based assay for the pharmaceutical industry.

## Supporting information

**S1 Dataset. Single-cells lactate measurements.**
(JNB)

**S2 Dataset. Warburg index determination.**
(JNB)

**S3 Dataset. Stable MDA-MB-231 cells.**
(JNB)

**S4 Dataset. Single-well detection of lactate accumulation.**
(JNB)

**S5 Dataset. Enhancement of lactate accumulation.**
(JNB)

**S6 Dataset. Z´-Factor determination.**
(JNB)

**S7 Dataset. Variation coefficient determination.**
(JNB)

**S8 Dataset. IC$_{50}$ for classical mitochondrial toxicants.**
(JNB)

**S9 Dataset. Pilot screening.**
(JNB)

**S10 Dataset. Stability of toxicity-induced lactate accumulation.**
(JNB)

## Author Contributions

**Conceptualization:** Luis Felipe Barros, Alejandro San Martín.

**Data curation:** Yasna Contreras-Baeza, Alejandro San Martín.

**Formal analysis:** Alejandro San Martín.

**Funding acquisition:** Luis Felipe Barros, Alejandro San Martín.

**Investigation:** Yasna Contreras-Baeza, Sebastián Ceballo, Robinson Arce-Molina, Pamela Y. Sandoval, Karin Alegría, Alejandro San Martín.

**Methodology:** Alejandro San Martín.

**Project administration:** Alejandro San Martín.

**Resources:** Alejandro San Martín.

**Supervision:** Alejandro San Martín.

**Validation:** Yasna Contreras-Baeza.

**Writing – original draft:** Luis Felipe Barros, Alejandro San Martín.

**Writing – review & editing:** Pamela Y. Sandoval, Luis Felipe Barros, Alejandro San Martín.

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
