## [Decision Letter · Decision Letter 0]

9 Sep 2019

PONE-D-19-20790

MitoToxy Assay: a novel cell-based method for the assessment of metabolic toxicity in a multiwell plate format using a lactate FRET nanosensor, Laconic

PLOS ONE

Dear Dr. San Martin,

Thank you for submitting your manuscript to PLOS ONE. After careful consideration, we feel that it has merit but does not fully meet PLOS ONE’s publication criteria as it currently stands. Therefore, we invite you to submit a revised version of the manuscript that addresses all the points raised by the reviewers.

We would appreciate receiving your revised manuscript by Oct 24 2019 11:59PM. To enhance the reproducibility of your results, we recommend that if applicable you deposit your laboratory protocols in protocols.io, where a protocol can be assigned its own identifier (DOI) such that it can be cited independently in the future. For instructions see: http://journals.plos.org/plosone/s/submission-guidelines#loc-laboratory-protocols

We look forward to receiving your revised manuscript.

Kind regards,

Mária A. Deli, M.D., Ph.D.

Academic Editor

PLOS ONE

Journal Requirements:

2. Please include your tables as part of your main manuscript and remove the individual files. Please note that supplementary tables should be uploaded as separate "supporting information" files.

Reviewers' comments:

Reviewer's Responses to Questions

**Comments to the Author**

1. Is the manuscript technically sound, and do the data support the conclusions?

Reviewer #1: Yes

Reviewer #2: Partly

2. Has the statistical analysis been performed appropriately and rigorously? 

Reviewer #1: Yes

Reviewer #2: No

3. Have the authors made all data underlying the findings in their manuscript fully available?

Reviewer #1: Yes

Reviewer #2: Yes

4. Is the manuscript presented in an intelligible fashion and written in standard English?

Reviewer #1: Yes

Reviewer #2: No

5. Review Comments to the Author

Reviewer #1: The current paper, written by Contreras-Baeza et al. describes a novel cell-based method for the assessment of metabolic toxicity during high-throughput drug screenings using a lactate FRET nanosensor, so called Laconic method, a measure of rapid lactate accumulation.

The manuscript is well-written. The Laconic assay provides a quick and sensitive measurement. Advantages and weaknesses of the method are well-detailed.

Table 1. provides a valuable comparison among the latest metabolic methods.

My only suggestion is that while the well-known mitochondrial inhibitors show a prompt response, not all compounds having such effects might show the response so rapidly. Therefore, I suggest to recommend using the assay at/after 30 mins (as it was done in Figure 8. and 9.) in order not to detect these compounds as false negatives.

Reviewer #2: The authors of the manuscript present a novel assay to detect potential mitochondrial dysfunction based upon a sensor designed to detect changes in lactate accumulation as a marker of glycolytic activation. The authors designed and performed experiments on a range of cells to inform the choice of which cell was most OXPHOS active and thus most suitable for insertion of the nanosensor. Once produced the cell model was characterised for utility using a range of classic mitochondrial toxins, hepatotoxins and negative controls.

Overall, the study describes a novel method which may find utility in the drug screening program. However, this reviewer has several comments upon the design of the study and interpretation of the results which the authors for the authors.

1) Today, the majority of drug companies use the glu/gal screen to detect mitochondrial toxicity (2 or 24 h), see PMID 25023361. However, this assay is only very briefly mentioned in the manuscript. This should be the main comparator for their work, as it is this that their assay would be replacing. However, as they are both based on the same physiological event, the switch to glycolysis, it is questionable of how much more useful this would be in terms of increasing specificity and selectivity.

2) The authors assess a panel of cells in order to select the candidate for sensor insertion. Their results show that MDA-MB-231 cells are more oxidative when assessed in galactose media. However, it does not appear that any of the other cells were assessed in galactose media, and so this is an unfair comparison. Importantly the aforementioned glu/gal assay is routinely done in HepG2 cells and is currently in place in most preclinical screens already as a screen for hepatic toxicity. Therefore, this reviewer believes that it is important to also assess WI index of HepG2 cells in galactose.

3) The figures lack any evidence of statistical analysis of whether changes in delta R are significant..

4) During assay refinement the authors demonstrate that the use a broad-spectrum inhibitor of MCT4 improves assay variability and sensitivity. Can they comment on whether the inclusion of such an inhibitor could effect drug dynamics?

5) In their assessment of how good their assay is (pg 20), the authors should consider their results in light of the Cmax. This would give an indication of whether any effects seen on the mitochondria would translate to clinical use. Furthermore, The authors use the phrase “false positives” in term of drugs which cause MMP changes but are considered relatively safe (pg 20, 509 – 512). This is important, as they should not consider this assay as an indicator of drug safety, this is probably too ambitious – in the absence of further information, it is only able to report on whether a drug affects the bioenergetic phenotype of a cell due to interactions with the mitochondrial ETC.

6) Some of the author’s comments do not adequately reflect the literature on the subject of screens to detect mitochondrial toxicity. For example (pg 20, line 499), publications have demonstrated mitochondrial toxicity of troglitazone at lower concentrations and short time-points (PMID 30096366 and 25746382).

7) The authors may like to comment on limitations of the use of this cell system to assess drug-induced metabolic toxicity, for example a lack of Phase 1 and Phase 2 metabolising enzymes. In addition mitochondrial toxicity often manifests over extended periods which may not be detected by their system.

6. PLOS authors have the option to publish the peer review history of their article (what does this mean?). If published, this will include your full peer review and any attached files.

Reviewer #1: No

Reviewer #2: No

---

## [Author Response · Author response to Decision Letter 0]

11 Oct 2019

Response to Reviewers

Reviewer #1: The current paper, written by Contreras-Baeza et al. describes a novel cell-based method for the assessment of metabolic toxicity during high-throughput drug screenings using a lactate FRET nanosensor, so called Laconic method, a measure of rapid lactate accumulation.

The manuscript is well-written. The Laconic assay provides a quick and sensitive measurement. Advantages and weaknesses of the method are well-detailed. Table 1. provides a valuable comparison among the latest metabolic methods.

Author: We thank the reviewer for his/her positive comments.

My only suggestion is that while the well-known mitochondrial inhibitors show a prompt response, not all compounds having such effects might show the response so rapidly. Therefore, I suggest to recommend using the assay at/after 30 mins (as it was done in Figure 8. and 9.) in order not to detect these compounds as false negatives.

Author: The point is well taken. The recommendation has been included in the Discussion section: Lines 530-533 ”Although well-known mitochondrial inhibitors elicited a prompt Laconic response (within 5 min), not all toxic compounds might show the response so rapidly. To minimize the frequency of false negatives results, we recommend measurements to be carried out at or after 30 minutes of exposure.”

Reviewer #2: The authors of the manuscript present a novel assay to detect potential mitochondrial dysfunction based upon a sensor designed to detect changes in lactate accumulation as a marker of glycolytic activation. The authors designed and performed experiments on a range of cells to inform the choice of which cell was most OXPHOS active and thus most suitable for insertion of the nanosensor. Once produced the cell model was characterized for utility using a range of classic mitochondrial toxins, hepatotoxins and negative controls.

Overall, the study describes a novel method which may find utility in the drug screening program. 

Author: We thank the reviewer for his/her positive appraisal of the article and the potential of the technique.

However, this reviewer has several comments upon the design of the study and interpretation of the results which the authors for the authors.

1) Today, the majority of drug companies use the glu/gal screen to detect mitochondrial toxicity (2 or 24 h), see PMID 25023361. However, this assay is only very briefly mentioned in the manuscript. This should be the main comparator for their work, as it is this that their assay would be replacing. However, as they are both based on the same physiological event, the switch to glycolysis, it is questionable of how much more useful this would be in terms of increasing specificity and selectivity.

Author: We thank the reviewer for directing our attention to the use of the glu/gal screen for drug discovery. The revised manuscript explains the importance of gluc/gal screens, which are typically based on measurement of oxygen, membrane potential and ATP. We are now citing PMID:25023361 and PMID:23147640, PMID:22689143, PMID: 17361016 and 21818751, which we deem most relevant to this paper. As it also uses galactose, the present method may be considered a variation of the glu/gal assay in which the readout is not oxygen consumption or cytosolic ATP levels but intracellular lactate. Although the present methods has advantages in terms of sensitivity, speed of detection and cost, we concur that it is not obvious that is better than oxygen/extracellular acidification detection in terms of specificity and selectivity. This is now discussed between lines 580-587, with reference to Table 1. 

2) The authors assess a panel of cells in order to select the candidate for sensor insertion. Their results show that MDA-MB-231 cells are more oxidative when assessed in galactose media. However, it does not appear that any of the other cells were assessed in galactose media, and so this is an unfair comparison. Importantly the aforementioned glu/gal assay is routinely done in HepG2 cells and is currently in place in most preclinical screens already as a screen for hepatic toxicity. Therefore, this reviewer believes that it is important to also assess WI index of HepG2 cells in galactose.

Author: New experiments were carried to assess WI in HepG2 in galactose as suggested. Galactose-fed HepG2 cells showed a low WI, similar that measured in MDA cells (Fig. 2D). However, their response to azide in terms of lactate accumulation was much weaker (Fig. 2C). These results are now described in lines 291-296.Therefore, whereas MDA cells are still the best option for toxicity detection using lactate levels, HepG2 cells could be approached with the WI, a more cumbersome protocol. This is commented in the Discussion section (lines 484-490). 

3) The figures lack any evidence of statistical analysis of whether changes in delta R are significant.

Author: Data were found to pass normality distribution and therefore statistical significance of the changes in ΔR was assessed using ANOVA. This is now indicated in Figures 9 and 10, and in Material and Methods, lines 232-235.

4) During assay refinement the authors demonstrate that the use a broad-spectrum inhibitor of MCT4 improves assay variability and sensitivity. Can they comment on whether the inclusion of such an inhibitor could affect drug dynamics?

Author: This is a valid point. A toxicant that does not have access to mitochondria because of interference by pCMBS may not be detected, generating a false negative. If hydrophilic compounds are tested, we recommend skipping pCMBS. To preserve sensitivity, the assay would have to be performed in triplicate. This is now discussed in lines 558-564.

5) In their assessment of how good their assay is (pg 20), the authors should consider their results in light of the Cmax. This would give an indication of whether any effects seen on the mitochondria would translate to clinical use. Furthermore, the authors use the phrase “false positives” in term of drugs which cause MMP changes but are considered relatively safe (pg 20, 509 – 512). This is important, as they should not consider this assay as an indicator of drug safety, this is probably too ambitious – in the absence of further information, it is only able to report on whether a drug affects the bioenergetic phenotype of a cell due to interactions with the mitochondrial ETC.

Author: We have added a paragraph that relates our findings with the respective values of the Cmax, Lines 524-527 “Our pilot screening detected the toxicity of ciglitazone, troglitazone, rosiglitazone and camptothecin at 5 µM, whereas the mitochondrial effects of terfenadine and flutamide were detected at 10 µM. As a reference, the maximum concentration of these drugs in plasma (Cmax) lies in the range of 3 to 10 µM (PMID:22987451, PMID:8792418, PMID:3986303, PMID:12235921 and PMID:2306725)”.

6) Some of the author’s comments do not adequately reflect the literature on the subject of screens to detect mitochondrial toxicity. For example (pg 20, line 499), publications have demonstrated mitochondrial toxicity of troglitazone at lower concentrations and short time-points (PMID 30096366 and 25746382).

Author: Thanks for bringing to our attention these papers. We have added PMID 30096366 and 25746382 to the reference section. Also, we have corrected the manuscript Lines 506-508 “The hitherto most sensitive detection of thiazolidinedione toxicity was obtained for troglitazone, which induced ATP depletion with an IC50 at 28 and 31,4 µM (PMID:30096366 and PMID:25746382).”

7) The authors may like to comment on limitations of the use of this cell system to assess drug-induced metabolic toxicity, for example a lack of Phase 1 and Phase 2 metabolizing enzymes. In addition mitochondrial toxicity often manifests over extended periods which may not be detected by their system.

Author: We concur. An assay based on cultured cells like the present one risks overlooking toxicants that are tissue specific. Not much is known about inter tissue mitochondrial variability. Interestingly, MDA-MB-231 cells do express Phase 1 and Phase 2 metabolizing enzymes PMID: 23873331. This is now mentioned in Lines 484-490. We agree that mitochondrial toxicity could manifest over extended periods which may not be detected by our assay. This is now mentioned in Lines 533-535. We are planning to test a variation of the present method for its ability to detect compounds whose toxicity develops over extended periods.

---

## [Decision Letter · Decision Letter 1]

16 Oct 2019

MitoToxy Assay: a novel cell-based method for the assessment of metabolic toxicity in a multiwell plate format using a lactate FRET nanosensor, Laconic

PONE-D-19-20790R1

Dear Dr. San Martin,

We are pleased to inform you that your manuscript has been judged scientifically suitable for publication and will be formally accepted for publication once it complies with all outstanding technical requirements.

With kind regards,

Mária A. Deli, M.D., Ph.D.

Academic Editor

PLOS ONE

Additional Editor Comments (optional):

Reviewers' comments:

Reviewer's Responses to Questions

**Comments to the Author**

1. If the authors have adequately addressed your comments raised in a previous round of review and you feel that this manuscript is now acceptable for publication, you may indicate that here to bypass the “Comments to the Author” section, enter your conflict of interest statement in the “Confidential to Editor” section, and submit your "Accept" recommendation.

Reviewer #1: All comments have been addressed

Reviewer #2: All comments have been addressed

2. Is the manuscript technically sound, and do the data support the conclusions?

Reviewer #1: Yes

Reviewer #2: Yes

3. Has the statistical analysis been performed appropriately and rigorously? 

Reviewer #1: Yes

Reviewer #2: Yes

4. Have the authors made all data underlying the findings in their manuscript fully available?

Reviewer #1: Yes

Reviewer #2: (No Response)

5. Is the manuscript presented in an intelligible fashion and written in standard English?

Reviewer #1: Yes

Reviewer #2: Yes

6. Review Comments to the Author

Reviewer #1: (No Response)

Reviewer #2: I thank the authors for the additions that they have made to the manuscript based upon my comments.

7. PLOS authors have the option to publish the peer review history of their article (what does this mean?). If published, this will include your full peer review and any attached files.

Reviewer #1: No

Reviewer #2: No

---

## [Editor Report · Acceptance letter]

23 Oct 2019

PONE-D-19-20790R1 

MitoToxy Assay: a novel cell-based method for the assessment of metabolic toxicity in a multiwell plate format using a lactate FRET nanosensor, Laconic 

Dear Dr. San Martín:

I am pleased to inform you that your manuscript has been deemed suitable for publication in PLOS ONE. Congratulations! Your manuscript is now with our production department. 

With kind regards,

on behalf of

Dr. Mária A. Deli 

Academic Editor

PLOS ONE